# Going Green (and Not Being Just More Pro-Social): Do Attitude and Personality Specifically Influence Pro-Environmental Behavior?

Jana S. Kesenheimer * and Tobias Greitemeyer

Institute for Psychology, University of Innsbruck, Innrain 52, 6020 Innsbruck, Austria;
tobias.greitemeyer@uibk.ac.at
* Correspondence: jana.kesenheimer@uibk.ac.at

**Abstract:** The current research examines the extent to which attitudes and personality traits are predictive of pro-environmental behavior (PEB). Concretely, we tested the relationship between pro-environmental attitudes, HEXACO personality traits, and actual PEB (donating potential prize money to a pro-environmental organization; $N = 257$). Additionally, we controlled for the influence of helping behavior (donating to a pro-social organization) by addressing whether attitudes and personality have a distinct impact on PEB or whether people are more likely to engage in PEB because they act more pro-socially in general. Analyses included correlations, multiple linear regressions, mediations, and partial correlations. Pro-environmental attitude had the most robust association with PEB and mediated the influence of openness to experiences and honesty–humility on PEB. Importantly, the relationship of pro-environmental attitudes and personality (openness to experiences and honesty–humility) with PEB was unaffected by the participant's helping behavior, suggesting that pro-environmental people mainly care about the environment and are not necessarily more pro-social in general.

**Keywords:** pro-environmental behavior; pro-social behavior; pro-environmental attitude; HEXACO personality





## 1. Introduction

Climate change is one of the most challenging issues facing humankind [1]. Interpersonal differences in environmental concerns have been of psychological scientific interest and result in the questions—who cares about ecology? ([2], p. 190) and what personality dimensions characterize a "pro-environmental individual"? ([3], p. 81). Indeed, it has been shown that personality, attitudes, as well as multiple demographics, such as political orientation, age, gender, social class, and education, influence pro-environmental behavior (PEB) [4]. When investigating the determinants of PEB, research findings have often been based on participants' self-reported behavior [2,3,5]. Even though this is common practice, self-reported measures have important limitations [6,7]. Lange and Dewitte [7], for example, argued that self-reports can be "shown at no cost" (p. 94) and do not have actual environmental consequences. In addition, they are often of a retrospective nature and therefore prone to bias. Thus, self-reported general behavior tends to reflect attitudes rather than being related to actual, observable behavior [8]. For example, when pro-environmental attitude, general self-reported, and recycling behavior were assessed [9], attitude and self-reported behavior highly correlated. However, only one-sixth of the participants who reported pro-recycling behavior in self-reports actually recycled a paper notice that they received four days later.

Because of these limitations of self-reported PEB, we assessed participants' actual PEB in the current study. Participants were asked at the end of the survey whether they would like to donate money to a pro-environmental organization. We examined the extent

to which pro-environmental attitudes and basic personality traits were predictive of the amount of money that was donated. Further, as detailed below, pro-environmental attitude can be described as a "precondition for pro-environmental behavior" ([10], p. 320). Hence, we expected pro-environmental attitude to mediate the relationship between personality traits and PEB. To put it differently, our reasoning was that personality is a distal determinant of PEB, whereas attitude is the proximal determinant and that attitude accounts for the impact of personality.

Finally, we addressed whether pro-environmental attitude and personality traits would have a distinct impact on PEB, or whether they would be predictive of other forms of pro-social behaviors as well (i.e., helping behavior). PEB and helping behavior can be seen as "a similar class of behavior" ([11], p. 1), as both are of communal value and can be subsumed under the umbrella term of pro-social behavior. Hence, it may be that people endorsing pro-environmental attitudes and/or scoring high on typical pro-social personality traits (e.g., agreeableness [3]) are more likely than others to engage in PEB, but only because they behave more pro-socially in general. If that is the case, attitudes and personality traits would not be specifically predictive of a person's PEB.

## 2. Previous Research on the Determinants of PEB

**Personality.** Humans differ in multiple aspects of their personality. Various personality models were developed to describe these differences. Becoming one dominant theory, five dimensions of personality are sufficient to approximately define an individual personality [12–14], namely, extraversion, agreeableness, conscientiousness, neuroticism, and openness. Extraversion describes talkative, assertive, and highly energetic people. The trait of agreeableness can be described with attributes such as being good-natured, cooperative, and trustful. Conscientiousness is characterized by being orderly, responsible, and dependable. Neuroticism is defined as a counterpart of emotional stability. Openness goes along with being intellectual, imaginative, and independent-minded. Based on this model, Lee and Ashton [15] further developed the big five taxonomy to six major dimensions of personality. The resulting model was named HEXACO, which is an acronym of the factors it includes: honesty–humility, emotionality, extraversion, agreeableness, conscientiousness, and openness to experiences. When comparing the HEXACO model to the big five taxonomy, the main difference is the added honesty–humility dimension [16]. To specify the dimension of honesty–humility, high levels are characterized by being sincere, fair, and unassuming. In contrast, lower levels are characterized as sly, greedy, and pretentious individuals [17].

Recently, Soutter, Bates, and Mõttus [18] found that honesty–humility and openness were the strongest predictors of pro-environmental attitude ($r = 0.22$ and $0.20$) and behavior ($r = 0.21$ and $0.25$), when investigating the HEXACO model and big five taxonomy in a meta-analysis. In addition, but to a lesser extent, agreeableness, conscientiousness, and extraversion were associated with pro-environmental attitude and behavior [18]. Other findings also revealed honesty–humility to be a significant determinant of PEB [19]. Additionally, Pavalache-Ilie and Cazan [19] found that not only honesty–humility predicted PEB, but also agreeableness, openness, a proactive personality, and pro-environmental attitude. In summary, honesty–humility and openness to experiences are especially related to PEB [10,18], whereas the other HEXACO personality traits are associated with PEB to a lesser extent [18].

**Pro-environmental attitude.** Also labeled as an "environmental concern", pro-environmental attitude is described as an orientation pattern that is stable across situations by being concerned about the natural environment [20,21]. Early works explained PEB as a linear function of environmental knowledge, leading to pro-environmental attitude, which in turn evolves into PEB (e.g., [22,23]). Recent findings suggest that pro-environmental attitude is related to PEB, but the relationship is far from perfect. There truly is "a gap between attitude and behavior in the environmental context" ([18], p. 922). According to Bamberg [20], a maximum of 10 percent variance of specific PEB can be explained by pro-

environmental attitude. Indeed, PEB is related to a variety of motives, including not only biospheric values and attitudes, but also motives of self-enhancement [23]. Nevertheless, although recent studies on PEB point to the importance of multiple other determinants, pro-environmental attitude is still a crucial predictor of PEB [24].

**The interplay of attitude and personality in predicting PEB.** Theoretically, personality traits should be distal determinants of PEB, whereas attitudes are the proximal determinant (e.g., [25,26]). For example, the assumption of values indirectly influencing behavior through a mediation of attitudes is the theoretical fundament of the "cognitive hierarchy model" based on Homer and Kahle's findings [25]. The model postulates a causal flow from more abstract cognitions and values (such as personality) to mid-range conditions (such as attitudes) to specific behavior (e.g., PEB). The model was successfully tested cross-culturally in predicting ecological behavior through altruistic and self-enhancement values mediated by environmental attitudes [27]. That is, pro-environmental attitude is assumed to act as a gateway to PEB, with personality traits (e.g., honesty–humility and openness to experience [10]) being supportive but subordinated.

Indeed, there is empirical evidence for the mediating effects of attitudes in determining PEB. Markowitz et al. [3] showed "that the effect of openness on pro-environmental behaviors was fully mediated by individuals' environmental attitudes and connection to nature" (p. 81). Similarly, the findings of Brick and Lewis [5] showed that attitudes toward nature mediated the effects of openness, conscientiousness, and extraversion on self-reported behavior to reduce greenhouse gas emissions. In the context of littering in Ibadan, Nigeria, Oluyinka [28] found that altruism and locus of control, both traits of personality, indirectly influenced self-reported littering behavior, mediated by a person's attitude towards littering. Other findings, also in Nigeria, showed that pro-environmental attitude significantly mediated the relationship between self-concept, environmental self-efficacy, and self-reported PEB [29]. Taken together, both theoretical reasoning and empirical evidence are supportive of the notion that specific personality traits are related to PEB, but this relationship is considerably reduced when taking the role of attitude into account. Understanding the interplay of attitude and personality is helpful in promoting PEB and ensuring sustainability. To increase pro-environmental behavior, enhancing a pro-environmental attitude, as the gateway to PEB, should then be of prime relevance for intervention.

**Methods of previous research.** Most of the previously outlined findings have been based on the participant's self-reported PEB. Although self-reports have some advantages, such as being low-cost, easily accessible in large sample sizes, and can target different forms of PEB, they also have several drawbacks [7]. As Gifford [30] aptly put it: "after all, to be blunt, not concern for the environment, not felt responsibility, not subjective norms, not attitude toward the behavior, not goals, and not even behavioral intentions solve environmental problems. Only actual behavior will bring a resolution" (p. 552). That is, self-reports may not necessarily reflect actual behavior. Indeed, the results of a meta-analysis by Kormos and Gifford [6] suggest an overlap of self-reported behavioral intentions and actual PEB of only 20%. Inconsistencies mainly arise due to recall biases, incorrect memory, and subjectivity in self-reports. Hence, it is crucial to measure objective PEB [30]. Thus, in the current study, we did not rely on self-reported general PEB, but asked for a willingness to donate potential prize money to pro-environmental organizations. Willingness to donate to environmental organizations has already been successfully used as an indicator for PEB in previous studies (e.g., [31–33]).

### 3. Previous Findings on the Relationship between Pro-Environmental and Pro-Social Behavior

Research has often investigated PEB through the lens of theoretical constructs of pro-social behavior, such as empathy and altruism [34]. Altruism, theoretically defined by the "division of labour and cooperation between genetically unrelated individuals in large groups" ([35], p. 785), has been shown to be related to PEB and even ranked second among biocentric and egoistic motives [36]. Interestingly, research has also found evidence for our

proposed mediating impact of pro-environmental attitudes in that the relationship between altruism and PEB was accounted for by pro-environmental attitudes [27,28]. In accordance, some forms of environmental engagement benefit not only the environment, but also helps others (and even oneself) [37]. For example, recycling benefits the environment and preserves resources, but also helps to keep the environment clean for oneself and others. Thus, PEB was also described as a "helping behavior" [38]. Unsurprisingly, people's empathy level is positively related to PEB and pro-environmental attitudes [39]. Investigating personality traits, Hilbig, Zettler, Moshagen, and Heydasch [10] found honesty–humility to be the most predictive HEXACO factor for (self-reported) PEB and, importantly, parts of its influence were predictive via pro-social value orientations (as an indicator of cooperativeness and enhancing joint outcomes in social dilemmas). In addition, Lee, Ashton, Choi, and Zachariassen [40] found that the relationship between pro-environmental attitudes and behavior is affected by people's connectedness to humanity (although this effect was weaker than the relationship between connectedness to nature and pro-environmental attitude and PEB).

Hence, it might be that people behave pro-environmentally just because they are more pro-social in general. As an example, the association of agreeableness, conscientiousness, and honesty–humility with being a "good citizen" and having the "tendency to be well socialized" served as an explanation for the link between pro-sociality and PEB (e.g., [3], p. 87). Therefore, pro-environmental individuals might not only behave pro-environmentally, but are also more likely to exhibit other forms of pro-social behavior, such as helping behavior, and it is therefore important to control for the extent to which people behave pro-socially. However, to our knowledge, so far, no empirical distinction has ever been made between environmentally relevant behavior on the one hand and other forms of pro-social behavior on the other within the same study. We therefore deemed it important to examine the relationship between pro-environmental attitudes and personality with PEB, while controlling for participant's general pro-social tendencies. To this end, we also assessed the participants' helping behavior (i.e., donating to a non-profit organization that was not related to the environment). This enabled us to examine whether personality traits and pro-environmental attitudes have a truly distinct impact on PEB.

## 4. The Present Investigation

In the current study, rather than assessing PEB through general self-reports, actual PEB was measured. By doing so, methodological disadvantages such as recall biases and subjectivity could be prevented. Additionally, the lack of research on the relationship between personality and attitude and PEB while controlling for the participant's pro-sociality (helping behavior, in particular) initiated the following research questions: What exactly is the relationship between personality, pro-environmental attitude, and actual PEB? Are personality and pro-environmental attitude specific determinants of PEB or are these relationships driven by a general tendency to behave pro-socially?

We examined the following hypotheses: in line with previous research [3,5,18,19], positive zero-level correlations (without considering covariates) were expected for PEB and the following personality traits: openness to experiences (I), honesty–humility (II), extraversion (III), conscientiousness (IV), and agreeableness (V). We further expected that pro-environmental attitudes would be predictive of PEB (VI) and that pro-environmental attitude would account for the relationship between personality traits and PEB (VII; in line with [3,5,25–27]).

As noted, we controlled for the participants' helping behavior. In addition, we included several covariates in the analyses that have been shown to be associated with PEB in previous research: political orientation, age, gender, social class, and education. For example, political orientation was shown to potentially account for a null effect of openness to experience on PEB [41]. In addition, PEB is known to be positively correlated with education and age, as well as with female gender, also influencing the association of envi-

ronmental attitude and PEB [42]. Because PEB could be motivated by "saving money" [43], we controlled for financial net income.

## 5. Methodology of the Present Study

**Participants.** A total of 273 individuals took part in an online study. The required number of participants was calculated a priori by an analysis of power [44]. It determined a target sample size of 262 with an expected effect size of $IpI = 0.20$ ($\alpha = 0.05$). In total, 15 participants had to be excluded, as they failed to follow the instruction to state a donation sum within the financial limit of the potential prize money (see below). The resulting valid cases included 181 female participants, 74 male, and 3 people who stated another gender. The participants' mean age was 27.2 years ($SD = 7.8$). The sample can be described as academic—127 participants stated having a high-school degree and 103 participants stated having a university degree as their highest educational level. A total of 70.0% of participants ($N = 180$) were students, and 22.2% ($N = 57$) were employed or self-employed. Their monthly financial net income ranged from no income at all up to 8000 euros, whereas 39.7% earned between 850 and 5000 euros per month. As the mean political orientation was 3.13 ($SD = 1.00$), stated on a seven-point Likert scale (higher values indicated a political right orientation), the sample's political orientation was slightly biased to the political left. Political orientation did not correlate with age, income, employment, or education.

**Materials and Methods.** Participants were invited via a university email newsletter. First, they answered demographic questions concerning their age, gender, financial net income, occupation, political orientation, and education. A questionnaire assessing pro-environmental attitude followed [45]. It included 12 items, providing statements about two guidelines: ecological modernization and social–economic transformation. The first means the aim to unite economic and ecological action. The second statement represents a holistic transformation to a post-fossil fuel society. Statements were balanced in terms of these topics and equally addressed affective, cognitive, and conative impacts. The items asked for the participant's agreement on these statements by a four-point Likert scale (1 "not at all" to 4 "entirely"). One sample item is: "Growth has natural limits that are already reached in our industrialized world". Originally, the questionnaire was part of a larger questionnaire, conducted in a study by the German Federal Environmental Agency. Its validity and reliability have proven to be of a high standard [45]. Next, participants filled out the HEXACO-PI-60 [46], stating their agreement on a five-point Likert scale (1 "not agree at all" to 5 "fully agree"). A sample item, corresponding openness to experiences (reversed), is: "I would be quite bored by a visit to an art gallery".

At the end of the survey, participants took part in a raffle with the chance to win 25 euros as a reward for their participation. They were asked if they would like to donate (part of) the money in case they won. Four organizations were presented to which the money could possibly be donated. One of the organizations had the goal of promoting the environment (Worldwide Fund for Nature), whereas the remaining organizations (Care Germany, World Hunger Help, and Doctors without Borders) also had pro-social goals but were not oriented towards the environment. Alternatively, participants could name any other organization of their choice. There was also an option to split the win of 25 euros between different organizations or to keep it, or parts of it, for themselves. Therefore, participants could show pro-environmental behavior (donations to the pro-environment organization), helping behavior (donations to the pro-social organizations), as well as egoistic behavior at the same time. After data collection was completed, one participant was drawn, and the chosen amount of money was donated.

## 6. Results

The open source software R, version 3.6.2 [47], was used to analyze the data. Arithmetic means were calculated for the HEXACO scales [46] and pro-environmental attitude [45]. Donation behavior was coded in two categories depending on the choice of a pro-social or pro-environmental organization. A total of 80.2% ($N = 207$) of all participants

indicated they would donate to one or more organizations. The average donation sum was 16.70 euros ($SD$ = 10.13). Means, standard deviations, reliability, and zero order correlations of the measured covariates, donation behavior, and pro-environmental attitude are reported in Table 1. Spearman correlations were chosen because the variables did not fulfill the assumption of normal distribution, as Shapiro–Wilk tests showed.

**Table 1.** Spearman's rank rho correlations matrix (zero-order correlations).

|  | M | SD | 1 | 2 | 3 | 4 | 5 | 6 | 7 | 8 |
|---|---|---|---|---|---|---|---|---|---|---|
| 1. Honesty–humility | 3.68 | 0.69 | 0.75 | | | | | | | |
| 2. Emotionality | 3.21 | 0.68 | 0.01 [−0.13, 0.14] | 0.72 | | | | | | |
| 3. Extraversion | 3.37 | 0.69 | 0.07 [−0.06, 0.18] | −0.09 [−0.21, 0.03] | 0.75 | | | | | |
| 4. Agreeableness | 3.17 | 0.58 | 0.13 * [0.00, 0.25] | −0.07 [−0.19, 0.06] | 0.10 [−0.03, 0.22] | 0.70 | | | | |
| 5. Conscientiousness | 3.76 | 0.61 | 0.13 * [0.01, 0.25] | 0.05 [−0.08, 0.18] | 0.00 [−0.12, 0.12] | −0.02 [−0.15, 0.11] | 0.77 | | | |
| 6. Openness to experiences | 3.63 | 0.65 | 0.12 [−0.00, 0.25] | −0.12 [−0.23, 0.01] | 0.12 [−0.01, 0.25] | 0.05 [−0.08, 0.16] | 0.05 [−0.09, 0.17] | 0.72 | | |
| 7. Pro-environmental attitude | 3.28 | 0.43 | 0.31 *** [0.19, 0.42] | 0.16 ** [0.04, 0.29] | 0.18 ** [0.06, 0.30] | 0.01 [−0.12, 0.12] | 0.03 [−0.10, 0.14] | 0.20 ** [0.08, 0.32] | 0.86 | |
| 8. Pro-environmental donation behavior | 5.68 | 7.95 | 0.17 ** [0.04, 0.29] | −0.05 [−0.17, 0.08] | 0.03 [−0.10, 0.15] | 0.04 [−0.08, 0.17] | 0.07 [−0.06, 0.19] | 0.15 * [0.02, 0.25] | 0.28 *** [0.17, 0.40] | |
| 9. Pro-social donation behavior | 11.02 | 9.80 | 0.14 * [0.03, 0.26] | 0.12 [−0.01, 0.23] | 0.11 [−0.02, 0.22] | 0.12 [−0.01, 0.24] | 0.01 [−0.12, 0.14] | −0.04 [−0.15, 0.08] | 0.02 [−0.10, 0.15] | −0.25 *** [−0.37, −0.13] |

Note. Confidence intervals (95%) are illustrated in brackets. Level of significance: * $p < 0.05$, ** $p < 0.01$, *** $p < 0.00$. The diagonal (grey) specifies the scales reliability (Cronbach's α).

As predicted in hypotheses I and II, openness to experiences and honesty–humility significantly correlated with PEB. Perhaps due to the dependence between donations to the pro-environment organization and the pro-social organizations, pro-environmental and pro-social donation amounts significantly negatively correlated. In contrast, extraversion, conscientiousness, and agreeableness were not related to PEB. Hence, hypotheses III to V did not receive support from the data. In contrast, pro-environmental attitude was positively related to PEB (hypothesis VI). Honesty–humility was the only significant predictor of pro-social behavior.

Next, multiple regression analyses were conducted to investigate personality traits and pro-environmental attitude as predictors of PEB. In line with our reasoning that personality is a distal determinant and attitude is the proximal determinant of PEB, only the HEXACO personality traits were included as predictors in the first step. The first-step analysis revealed honesty–humility, $t(250) = 2.00$, $p = 0.047$, and openness to experiences $t(250) = 2.16$, $p = 0.032$, to be significant predictors of PEB. This regression, including the six personality dimensions, yielded a multiple $R^2$ value of 0.05. In the second step, environmental attitude, as well as covariates (age, gender: men vs. women, political orientation, education, and financial net income), were included in the regression model. Pro-environmental attitude was a significant predictor, $t(242) = 3.69$, $p < 0.001$, as well as gender, $t(242) = 2.64$, $p = 0.009$, with men donating more ($M = 7.53$, $SD = 9.51$) than women ($M = 4.94$, $SD = 7.14$; $p = 0.037$). No other significant effects were found. In total, the regression model explained 13.3% (multiple $R^2$) of variance.

We then tested hypothesis VII that pro-environmental attitude accounts for the relationship between personality traits and PEB. A sample of 500-bootstrap causal mediation analyses showed a significant full mediation effect regarding the relationship of honesty–humility with PEB, as well as a partial mediation effect for the relationship of openness to experiences with PEB. Both findings are illustrated in Figures 1 and 2. Moreover, even though the direct relationship between PEB and extraversion and emotionality, respectively,

were not significant, the indirect effect via pro-environmental attitude was significant. The exact statistics can be obtained from the first author upon request.

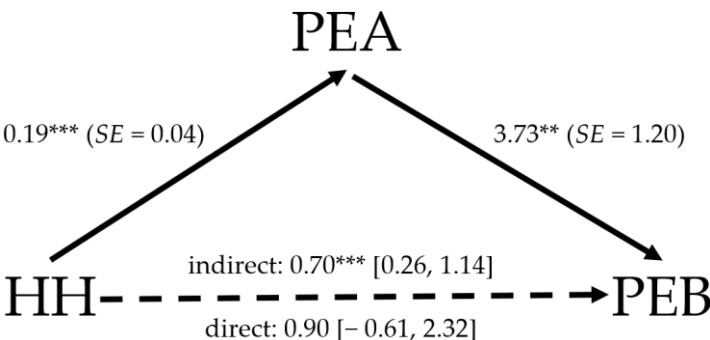

**Figure 1.** 500-sample bootstrap mediation analysis on honesty–humility (HH), pro-environmental behavior (PEB), and pro-environmental attitude (PEA). (Unstandardized) $\beta$-weights and 95% confidence intervals are illustrated. The indirect effect reflects the average causal mediation effect [total effect—direct effect]. The total effect of the mediation analysis was significant ($\beta$ = 1.61 [0.21, 3.17], $p$ = 0.024). Level of significance: ** $p < 0.01$, *** $p < 0.00$.

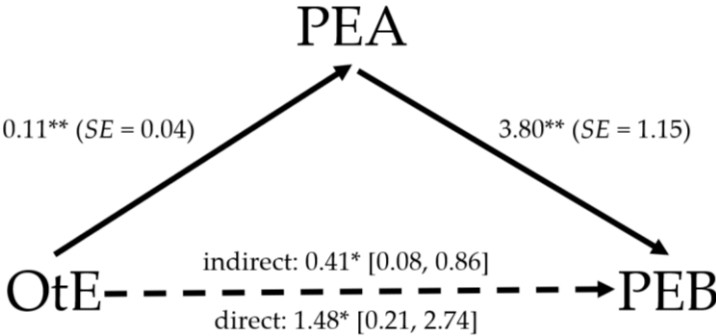

**Figure 2.** 500-sample bootstrap mediation analysis on openness to experiences (OtE), pro-environmental behavior (PEB), and pro-environmental attitude (PEA). (Unstandardized) $\beta$-weights and 95% confidence intervals are illustrated. The indirect effect reflects the average causal mediation effect [total effect—direct effect]. The total effect of the mediation analysis was significant ($\beta$ = 1.89 [0.65, 3.20], $p < 0.001$). Level of significance: * $p < 0.05$, ** $p < 0.01$.

Finally, our aim was to investigate the role of people's general pro-sociality in the determination of PEB. Partial Spearman correlations, controlling for helping behavior, showed that the relation of PEB to personality (honesty–humility: $r$ = 0.21, $p < 0.001$; openness to experiences: $r$ = 0.14, $p$ = 0.024) and pro-environmental attitude ($r$ = 0.30, $p < 0.001$) remained significant. Hence, the relation of personality traits and attitude to PEB was independent of the participants' level of pro-sociality.

## 7. Discussion

The aim of the present study was to extend knowledge about the relation of personality and attitude to actual PEB and to examine whether these relationships hold when taking another form of pro-social (i.e., helping) behavior into account. As in previous research, honesty–humility and openness to experiences were especially related to PEB [10,18]. Importantly, however, the present results highlight the predominant role of pro-environmental attitude compared to personality traits in determining PEB. In line with the cognitive hierarchy model [25], attitude significantly mediated the relationship between personality traits and PEB, specifically honesty-humility and openness to experiences, respectively. This mediation effect of attitude is in line with our hypotheses and previous research that claimed pro-environmental attitude to be a precondition for PEB [3,5,25–27].

Regarding honesty–humility, correlational analyses showed relations to both pro-social and pro-environmental behavior. Moral concerns about the welfare of strangers beyond one's in-group characterize honesty–humility [35]. In fact, Hilbig, Glöckner, and Zettler [48] found that the HEXACO dimension honesty–humility is a crucial predictor of pro-social behavior. We also found that out of the HEXACO personality traits, only honesty–humility significantly correlated with helping behavior. Importantly, PEB was not only related to honesty–humility, but also to pro-environmental attitudes as well, and the relation of honesty–humility to PEB was fully explained by environmental attitude. It is noteworthy that the current findings that are based on the observation of actual pro-environmental and helping behavior are in line with previous research using general self-reports of PEB [3,5,18,19].

As with honesty–humility, openness to experiences was related to PEB and the effect was mediated by differences in pro-environmental attitudes. This is in agreement with the definition of openness to experiences by the appreciation of natural wonders, by actively seeking new solutions to problems and being receptive to ideas that might seem strange or radical [49]. PEB often involves the willingness to change habits, behave differently than others and against personal or social traditions. In addition, the appreciation of nature is likely to be fundamental to protect it.

Contrary to our hypotheses, besides honesty–humility and openness to experiences, no other HEXACO dimensions [46] correlated with PEB. Some previous studies [3,5,19] did find significant correlations for extraversion, conscientiousness, and agreeableness, however, previous findings were inconsistent, especially when relying on self-reported PEB (e.g., [50]). Indeed, whereas honesty–humility and openness to experiences were consistently positively related to PEB in previous research [10,18], other HEXACO personality traits were associated with PEB to a lesser extent and considerably varied across studies [18]. Compared to previous research, the current study was not based on retrospective general self-reports of PEB. As noted in the introduction, attitudes and behavior align when investigating general self-reports [8]. Accordingly, the relationship between personality traits and PEB could be more pronounced when both are assessed using general self-reports.

Interestingly, controlling for the participant's level of helping behavior did not affect the direct relation of openness to experiences, honesty-humility, or environmental attitude to PEB. Thus, the determination of PEB was independent of the participant's general pro-sociality. This outlines the importance of clearly distinguishing pro-environmental and other forms of pro-social behavior, even though they are theoretically explained by similar models (e.g., [34]).

**Implications**. This present study has important theoretical and practical implications. On one hand, it addressed the interplay between multiple determinants of PEB and other forms of pro-social behavior. Although both behaviors overlap in terms of their relationship with honesty–humility, openness to experiences and pro-environmental attitudes were related to PEB, but they were not predictive of helping behavior. Common explanations for pro-social behavior in general and PEB in particular are often expected to be similar because people just aim to "do the right thing", and therefore need to be differentiated. In addition, the present finding that attitudes account for the impact of personality on PEB supports the cognitive hierarchy model [25] that pro-environmental attitude serves as the proximal determinant of PEB.

With respect to the sustainable development goals (SDGs; [51]), which provide a global vision of a socially, ecologically, and economically sustainable future, the present research contributes to the basic understanding of how PEB is determined. By understanding how environmental behavior evolves, "context-specific solutions" can be contributed ([52], p. 1486). For example, SDG goal 12 addresses responsible consumption and SDG goal 13 addresses climate action—both on an individual customer's level, as well as for production and supply chains [53]. In both cases, human "green" decision-making,

which depends on their environmental attitudes as the present study shows, is crucial for a sustainable development.

Thus, with regard to practical implications, to promote PEB, interventions should not target people's pro-social motives such as altruism, as people do not appear to be going green because of their pro-sociality. In addition, people are not going green in the first place because their personality makes them inclined to do so. Even though honesty–humility and openness to experiences were associated with PEB, interventions targeting people's pro-environmental attitudes seem to be more promising. For example, reading environmental literature has been shown to predict pro-environmental attitude and environmentally responsible behavior [54]. Nature-based online and book texts could therefore be helpful to promote PEB in school and leisure. For example, a smartphone-based intervention [55] providing nature-based literature might be helpful. From a psychological viewpoint, such interventions to promote pro-environmental attitudes could help to achieve sustainable development goals, especially those that emphasize "the need to improve energy efficiency, increase the share of clean and renewable energies and improve sustainable consumption patterns worldwide" (SDG 12; [56], p. 1).

**Limitations and future research.** As we found significant gender differences in pro-environmental attitude and PEB, and given the large gender imbalance in the present sample, further research addressing the impact of gender would be desirable. Another limitation is that findings might be limited to academic, highly educated, and pro-environmentally aware people (as most of our participants were). A bigger variation in demographics and pro-environmental attitude might even strengthen the correlations found in this study. Another limitation concerns the maximum amount of 25 euros that participants could have donated and that forced a decision to donate to either pro-environmental or pro-social organizations. In fact, PEB and pro-social behavior were negatively correlated, which could be a consequence of how we assessed both. Future research may therefore employ independent measurements of PEB and pro-social behavior.

Importantly, the cross-sectional design precludes strong causal conclusions. Nevertheless, our study provided an opportunity to falsify the predictions of the causal theory that inspired it. Because the results supported the theoretical predictions, the study adds support to the general theory (i.e., the cognitive hierarchy model that personality is related to attitudes, which in turn is related to actual behavior). Nonetheless, additional studies using a longitudinal design would be welcome.

Further research could address the possibility that correlations of personality traits and PEB in previous research simply evolved due to the general assessment of both. A next goal is to not only observe actual PEB, but also personality traits without relying on self-reports. For example, the latter could be observed by using peer reports. Further, as the observation of one single behavior might not reflect actual behavior with high reliability, multiple observations of actual behaviors are desirable. It is also important that the measure of PEB in the present study has no actual consequences for most of the participants, because the actual donation amount was eventually determined by only one of them.

## 8. Conclusions

Retrospective self-reports of PEB are known to have several limitations. As attitudes and behavior are proximate when observing general self-reports [8], so might personality traits and behavior. Moreover, although research has often investigated PEB through the lens of theoretical constructs of pro-social behavior, so far, no empirical distinction has ever been made between environmentally relevant behavior on the one hand and other forms of pro-social behavior on the other within the same study. Thus, in the present research, participants had the opportunity to show pro-environmental, helping, or egoistic behavior (by deciding to donate money to pro-environmental or pro-social humanistic organizations). We investigated if personality and pro-environmental attitude are specific determinants of PEB or if these relationships are driven by a general tendency to behave pro-socially.

Indeed, we found evidence that the determination of PEB is unaffected by people's helping behavior. Although both PEB and helping behavior go along with the honesty–humility personality trait, PEB was additionally predicted by openness to experiences and, more proximally, pro-environmental attitude. Most importantly, the present findings showed that attitude accounts for the impact of personality, in line with the cognitive hierarchy model [25].

To sum up, this research suggests that pro-environmental attitude serves as the proximal determinant of PEB. Future research that addresses the limitations of the present study would be welcome, such as observing pro-environmental attitude, pro-social behavior, and PEB by using a longitudinal design. Regarding practical implications, and in line with the present findings, to increase pro-environmental behavior, enhancing a pro-environmental attitude, as the gateway to PEB, should be of prime relevance for interventions to prevent further impacts of climate crisis. To achieve the sustainable development goals [51], interventions on individual environmental decision-making should therefore not appeal to people's altruistic side but should specifically address their care for the environment.

**Author Contributions:** Conceptualization, T.G. and J.S.K.; methodology, T.G.; formal analysis, J.S.K.; data collection and curation, T.G.; writing—original draft preparation, J.S.K.; writing—review and editing, T.G. and J.S.K. All authors have read and agreed to the published version of the manuscript.

**Funding:** This research received no external funding.

**Institutional Review Board Statement:** The study was conducted according to the guidelines of the Declaration of Helsinki. According to the Ethics Committee of the University of Innsbruck, no ethical approval was required for this study as no identifiable human data was gathered and no interventions took place in the context of this study.

**Informed Consent Statement:** Informed consent was obtained from all subjects involved in the study.

**Data Availability Statement:** Data can be obtained from the first author upon reasonable request.

**Acknowledgments:** We thank Sabrina Huck for her help in data collection.

**Conflicts of Interest:** The authors declare no conflict of interest. No ethical issues have been found to apply.

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
