# Peer review of "Going Green (and Not Being Just More Pro-Social): Do Attitude and Personality Specifically Influence Pro-Environmental Behavior?"

_sustainability, doi:10.3390/su13063560_

Round 1

Reviewer 1 Report

Review of Sustainability-1159947, [revised manuscript] “Going green (and not being just more pro-social): How attitude and personality are related to pro-environmental and pro-social behavior”

In my opinion, the authors have done a good job in addressing the points of my original review.  I don’t have any further recommendations for revision.

Author Response

Thank you again for your very helpful advice!

Reviewer 2 Report

The manuscript is much improved. However, I have a few suggestions as follows: 

1. The manuscript is far unbalanced. The introductory material and the description of the methodology is sufficiently detailed, however results, discussion are really short furthermore moreover, there are almost no  conclusions section!
2. In the discussion section, I propose to emphasize  how the research results fit into the issues of sustainable development and SDGs. 

    I suggest using the following sources:

- Smith, M. S., Cook, C., Sokona, Y., Elmqvist, T., Fukushi, K., Broadgate, W., & Jarzebski, M. P. (2018). Advancing sustainability science for the SDGs. Sustainability science, 13(6), 1483-1487.

- Zimon D., Tyan J., Sroufe R. (2020), Drivers of sustainable supply chain management: practices to alignment with un sustainable development goals, International Journal for Quality Research 14(1) 219–236.

- Fonseca, L.M.; Domingues, J.P.; Dima, A.M. Mapping the Sustainable Development Goals Relationships. Sustainability 2020, 12, 3359

etc. 

3.     Talking about the conclusion section, it should be the last section of the paper. It should contain three parts: a. summary of your research, main findings and defining the contribution and (obligatory) managerial/empirical implications, b. research limitations, c. future research directions.
    Conclusions need to be expanded in order to provide more references to sustainability.

I hope the author can work on these suggestions.

All the best!

Author Response

Thank you again for your very helpful comments. Below we provide our responses to each of the points that were raised; our responses are marked with an asterisk. Please feel free to contact us if you have any questions about this resubmission. Thank you again. We look forward to hearing from you.

The manuscript is much improved. However, I have a few suggestions as follows:
1. The manuscript is far unbalanced. The introductory material and the description of the methodology is sufficiently detailed, however results, discussion are really short furthermore moreover, there are almost no conclusions section!
* We agree with the reviewer that the discussion and the conclusion parts were relatively short. In our revision, we addressed the other points that were raised (see below). The discussion and conclusion section are now longer (pp. 8-10).
2. In the discussion section, I propose to emphasize how the research results fit into the issues of sustainable development and SDGs.
I suggest using the following sources:
- Smith, M. S., Cook, C., Sokona, Y., Elmqvist, T., Fukushi, K., Broadgate, W., & Jarzebski, M. P. (2018). Advancing sustainability science for the SDGs. Sustainability science, 13(6), 1483-1487.
- Zimon D., Tyan J., Sroufe R. (2020), Drivers of sustainable supply chain management: practices to alignment with un sustainable development goals, International Journal for Quality Research 14(1) 219–236.
- Fonseca, L.M.; Domingues, J.P.; Dima, A.M. Mapping the Sustainable Development Goals Relationships. Sustainability 2020, 12, 3359
etc.
* Thank you for alerting us to these references. We now refer to all of them in a new paragraph of the discussion section (lines 370-378), and rewrote other parts of the discussion (lines 388-392). In addition, we added information about the SDG’s in the conclusion (lines 442-445). You can find these new citations numbered by 51, 52, 53 and 56.
3. Talking about the conclusion section, it should be the last section of the paper. It should contain three parts: a. summary of your research, main findings and defining the contribution and (obligatory) managerial/empirical implications, b. research limitations, c. future research directions. Conclusions need to be expanded in order to provide more references to sustainability.
* As suggested by the reviewer, we rewrote the conclusion by summing up main findings, implications, limitations, future directions, and added a final conclusion (lines 420-445). It is the last section of the manuscript and now addresses all the points that were raised by summing up previous outlined parts of the discussion section.
I hope the author can work on these suggestions.
All the best!
* Thank you for your help to further improve our manuscript!

This manuscript is a resubmission of an earlier submission. The following is a list of the peer review reports and author responses from that submission.

Round 1

Reviewer 1 Report

Review of Sustainability-1097258, “Going green (and not being just more pro-social): How attitude and personality are related to pro-environmental and pro-social behavior”

This manuscript examines the associations of self-report personality characteristics (specifically, the HEXACO factors), pro-environmental attitudes, pro-environmental behaviours, and pro-social behaviours in an online sample of German students and other adults.  The pro-environmental and pro-social behaviour variables were assessed in terms of respondents’ decisions as to how to they would allocate money.  The results showed that Honesty-Humility and Openness were both modestly associated with pro-environmental behaviours, with both associations partly mediated by pro-environmental attitudes (which were more strongly related to pro-environmental behaviours).  Honesty-Humility also predicted pro-social behaviours, which were somewhat negatively correlated with pro-environmental behaviours.  (This negative correlation reflects the forced-choice format of the money allocation decision, whereby respondents could divide money between self, pro-social organization, and pro-environmental organization.)

I think that this manuscript addresses an interesting topic using some relevant data, and the research generally seems to have been conducted and reported well.  In my opinion, it should be published after some revisions are incorporated.  I’ve listed some comments below.

  1. p. 2, line 70: it would be better to say that the Big Five and HEXACO factor are “mostly similar”, because there are some differences between HEXACO and Big Five Agreeableness and between HEXACO Emotionality and Big Five Neuroticism

  1. p. 3, line 128: the phrase “were employed, occupational independently or civil servants” was difficult to understand; perhaps the authors meant “were employed by a private company, were self-employed, or were civil servants”   (I added “by a private company” because the authors elsewhere specified civil servants, who are employed)

  1. Figures 1 and 2: I believe that these are unstandardized B weights rather than beta weights (which are standardized); also, I notice that the products of the paths involving PEA are not quite equal to the indirect paths between personality and PEB, so these values should be checked

  1. p. 6, line 215: the phrasing here is ambiguous as to which variable was being correlated with which others; it should say “relations of PEB with personality and pro-environmental attitude” (note the “with”)

  1. p. 6, line 216: when I use the values in Table 1, I can’t recover the same partial correlations as those listed by the authors, and I don’t think that the differences are due to rounding error; it would be good to check the numbers (I think the value for Honesty-Humility is right, but I get a partial correlation of .16 for Openness (not .14), and about .28 for pro-environmental attitudes (not .39))

  1. p. 6, line 221: it would be clearer to say “relations of personality and attitudes with PEB”

  1. p. 7, line 255-256: I could not understand the sentence “As previous findings showed…”, and I think it would be better to re-write it entirely

  1. The authors should cite the studies of personality and environmental behaviours by Hilbig, Zettler, Moshagen, and Heydasch (2013) and by Lee, Ashton, Choi, and Zachariassen (2015), and they should also cite the recent meta-analysis by Soutter, Bates, and Mõttus (2020).

Reviewer 2 Report

In the manuscript, the authors presented "Going Green (and Not Being just More Pro-Social): How Attitude and Personality are Related to Pro-Environmental and Pro-Social Behavior". The manuscript is only 9 pages long with a reference list. In the manuscript, the authors presented an important point, although the manuscript has some drawbacks:

  1. Intruduction - in the introduction, please write about the research gap. Too short introduction.
  2. Results - the manuscript has been sent to Sustainability. Please refer to sustainable development (This is missing).
  3. Conclusions - conclusions are too short, only 1 paragraph. In your applications, please also answer the following questions: what are the directions for the future? what is new to this manuscript?
  4. References - please expand the literature list, only 29 items.

Reviewer 3 Report

Thank you for the opportunity to read this interesting paper. I think the paper is interesting but unfortunately, at theoretically and methodologically the paper is still very underdeveloped. Below you can find my detailed comments about how the paper could be developed further.

1. INTRODUCTION: The introduction is very weak, it doesn't support the research topic and goal: an exhaustive explanation of research motivations is fundamental in a scientific paper. I think that the point of view of this research could be an interesting novelty for the scientists, but at the moment, the paper is not able to demonstrate the value of the research Some methodological insights should be added to the Introduction. Please review the literature gap analysis, assuring an internal coherence between research topic, emerging findings, knowledge gaps and research goals (and then, research questions).
2. Emphasize your goals and research questions. The research objectives and methodology should be better explained and motivated.
3. The literature review is rather limited. The literature review should be broader.
4. Try to introduce your methodologies in introduction section itself, which makes the readers more familiar with these tools.
5. Authors should better develop a section with methodology.
6. Overall, try to provide sufficient validation regarding the novelty of this research along with beneficial.

Good Luck!